# Metabolic Effects of an Oral Glucose Tolerance Test Compared to the Mixed Meal Tolerance Tests: A Narrative Review

**DOI:** 10.3390/nu14102032

**Published:** 2022-05-12

**Authors:** Marlene Lages, Renata Barros, Pedro Moreira, Maria P. Guarino

**Affiliations:** 1ciTechCare—Center for Innovative Care and Health Technology, Polytechnic of Leiria, 2410-541 Leiria, Portugal; marlene.c.lages@ipleiria.pt; 2Faculty of Nutrition and Food Science, University of Porto, 4150-180 Porto, Portugal; renatabarros@fcna.up.pt (R.B.); pedromoreira@fcna.up.pt (P.M.); 3EPIUnit—Instituto de Saude Publica, Universidade do Porto, 4200-450 Porto, Portugal; 4Laboratorio Para a Investigação Integrativa e Translacional em Saude Populacional (ITR), Portugal Centre in Physical Activity, Health and Leisure, University of Porto, 4200-450 Porto, Portugal; 5School of Health Sciences, Polytechnic of Leiria, 2411-901 Leiria, Portugal

**Keywords:** diabetes, diagnosis, insulin resistance, fasting plasma glucose, impaired fasting glycemia, impaired glucose tolerance

## Abstract

The oral glucose tolerance test (OGTT) is recommended for assessing abnormalities in glucose homeostasis. Recognised as the gold standard test for diagnosing diabetes, the OGTT provides useful information about glucose tolerance. However, it does not replicate the process of absorption and digestion of complex foods, such as that which occurs with a mixed meal tolerance test (MMTT), an alternative that is still not well explored in the diagnosis of metabolic alterations. The MMTT could be an asset in detecting glucose homeostasis disorders, including diabetes since it has more similarities to the common dietary pattern, allowing early detection of subtle changes in metabolic homeostasis in response to combined nutrients. This alternative has the advantage of being more tolerable and pleasant to patients since it induces a more gradual increase in blood glucose, thus reducing the risk of rebound hypoglycemia and other related complications. The present article reviewed the clinical data available regarding the possibility of screening or diagnosing altered glucose homeostasis, including type 2 diabetes mellitus, with the MMTT.

## 1. Introduction

Diabetes mellitus (DM) includes a cluster of metabolic conditions defined by hyperglycemia that can result from insufficient insulin secretion, defects in insulin action, or both [1]. The global prevalence of DM continues to increase, placing an ever-increasing burden on healthcare systems due to the disease and its complications [2]. Currently, 351.7 million people aged between 20 and 64 years have diabetes, whether it is diagnosed or undiagnosed diabetes. By 2030, it is expected that this number will increase to 417.3 million and by 2045 it will reach 486.1 million people [3]. Globally, 213.9 million of the 463 million, or half of the adults that live with diabetes, do not know that they have this disease [3]. In Portugal, in 2018, the estimated prevalence of diabetes was 13.6% in the population aged between 20 and 79 years old. According to these numbers, more than 1 million Portuguese in this age group have diabetes, of which 56% have already been diagnosed and 44% have not yet been diagnosed [4].

Early screening and detection are of crucial importance since continued undiagnosed diabetes increases the risk of having diabetes-related complications [5]. Most cases of DM-associated morbidity and mortality are caused by macrovascular and microvascular complications, some associated with late diagnosis [6]. Diagnosis and classification of diabetes are important for determining the more effective treatment, whether this is based on lifestyle changes, pharmacological therapy, or a combination of both therapeutic approaches.

A report from the World Health Organization (WHO) and the International Diabetes Federation considered the OGTT as the gold standard test used to diagnose diabetes [7]. Despite providing useful information about glucose tolerance, it does not provide information about insulin sensitivity or resistance mostly due to its main limitation, which is that it does not mimic the physiological postprandial metabolism.

The standardized mixed meal has been pointed out as a substitute for the glucose solution administered in the OGTT. In addition, the administration of 75 g of glucose may cause unpleasant symptoms, such as nausea, vomiting, diarrhoea, bloating and anxiety [8,9,10]. The metabolic feedback to a mixed meal is a better indication of beta-cell function under normal daily life conditions compared with the standard OGTT since the mixed meal contains proteins and fatty acids, which are components that can stimulate insulin secretion [11,12,13]. In the literature, some studies compare blood glucose levels and insulin sensitivity and resistance during the OGTT and a mixed meal tolerance test (MMTT) [14]. However, the meals used in these clinical studies are diverse and sometimes have a different nutritional composition, leading to heterogeneous results. To the best of our knowledge, there are no previous review articles published that directly compare the results of an OGTT to the different mixed meal tests. Thus, the present review aimed to assess and compare the metabolic effects of the OGTT and mixed meal tests in clinical studies and provide a comprehensive overview of this area of research.

## 2. Materials and Methods

The search was conducted between October and November 2021 using the Scopus database. MeSH terms were applied when possible, and different search strategies were used combining the following keywords: “oral glucose tolerance test”, “mixed meal tolerance test”, “meal test”, “meal tolerance test”, “diabetes”, and “gestational diabetes”.

The search strategies had no date restrictions and included articles published in English and Portuguese. The exclusion criteria were: protocols, letters, commentaries, and studies that were not carried out within the scope of this review, including studies conducted in animals models and studies where the participants, or at least a subsample of participants, did not perform both the OGTT and MMTT to allow comparison. The date last searched was 31 December 2021. Hand searching and references from the extracted articles were also consulted. After selecting the original articles to include and discuss in the present review, the information about their results was synthesized in a table format, including the study design, characteristics of the participants, index test information (sensitivity, specificity, positive predictive value (PPV), negative predictive value (NPV)), and 2-h glucose correlation for the OGTT and MMTT whenever this was available.

## 3. Standard Methods for the Diagnosis of Diabetes

The diagnosis of diabetes is based on plasma glucose criteria, which include the value of fasting plasma glucose (FPG) or the 2-h plasma glucose (2-h PG) value during a 75 g OGTT, or glycated haemoglobin A1c (HbA1c) criteria [1]. The American Diabetes Association (ADA) considers that HbA1c values above 6.5% diagnose diabetes and are considered a significant marker to develop long-term diabetes complications. The use of HbA1c as a clinical endpoint has the advantage that it can be performed without the need for a fasting period, contrary to the FPG or the OGTT. Additionally, this biomarker has greater preanalytical stability and is an indirect measure of mean blood glucose values over the past three months, which eliminates day-to-day variability as a confounding factor in the assessment of impaired glucose tolerance (IGT) or impaired fasting glucose (IFG) [15]. Nonetheless, the use of a single biomarker to detect glucose disturbances has inherent limitations, including lower sensitivity and specificity, and inaccuracy since HbA1c can be affected by ethnicity, haemoglobin variants, and other clinical conditions [16,17,18].

Fasting plasma glucose is a method with sensitivity but has poor reproducibility. To measure this biomarker, it is required that patients have fasted for at least 8 h before the blood sample collection, and the value can still be affected by short-term factors such as stress and exercise [19,20]. The current recommendations for diabetes diagnosis are established based on studies that measure plasma glucose in blood samples collected in the morning following an overnight fast of a minimum of 8 h. However, some patients may perform this test in the afternoon after an 8 h day fast and, since plasma glucose values are elevated in the morning, it is uncertain if FPG criteria should remain the same for this situation [21,22]. Additionally, when FPG values are used isolated for diagnosis of diabetes, people with an FPG below 126 mg/dL may be misdiagnosed because results from this test can show a discordance from the 2-h PG [23,24,25]. Additionally, the degree of hyperglycemia changes over time and may present as glucose tolerance abnormalities without reaching the criteria for diabetes, depending on the time from diagnosis and progression of the disease process. Thus, IFG is identified as having FPG levels from 100 mg/dL to 125 mg/dL and IGT as 2-h values in the OGTT of 140 mg/dL to 199 mg/dL [26].

The OGTT is the reference method to diagnose type 2 diabetes mellitus (T2DM) and to define the categories of glucose intolerance that result from higher-than-normal blood glucose levels in the absence of frank diabetes mellitus [26]. The cut-off points allow detecting the intermediate conditions between normal glucose tolerance and diabetes, namely, IGT and IFG [26]. This test was standardized by the administration of a glucose load that contains the equivalent of 75 g of sugar dissolved in water. To screen for diabetes, the 2-h PG after the OGTT and the FPG are more accurate when there is a discordance between the HbA1c and glucose value [26,27].

## 4. Oral Glucose Tolerance Test and Mixed Meal Tolerance Tests

The OGTT and the MMTT are generally used in clinical research related to metabolism as well as diabetes drug development. These tests provide an integrated assessment of the β-cell response to an insulin secretory stimulus that comprises activation of the incretin axis. The main incretin hormones involved in this process are glucagon-like peptide 1, secreted by the L cells, and glucose-dependent insulinotropic peptide, secreted by the K cells. The activation of these incretins includes enhancement of glucose-stimulated insulin secretion from islet β-cells in combination with reduced glucagon release from α-cells [28]. However, a mixed meal is a more physiologic stimulus to insulin secretion because β-cells are also responsive to certain amino acids and fatty acids in addition to glucose [29,30].

The OGTT may reveal both glucose and insulin secretion/action disturbances, but it is important to point out that the liquid glucose can be quickly absorbed, causing an early release of insulin. This can cause false reactive hypoglycemia associated with adverse epigastric symptoms, which do not replicate the usual glucose excursion and insulin responses in daily life conditions. From a clinical perspective, the incidence of reactive hypoglycemia is important. There are reports of hypoglycemia-induced ischemic electrocardiogram changes [31,32], which can indicate a need for caution when performing an OGTT in patients with coronary heart disease. Reactive hypoglycemia can also be challenging in dumping syndrome patients since the liquid glucose promotes rapid gastric emptying [33,34]. Considering this, the exchange of the traditional glucose drink with a standard mixed meal could be beneficial to patients and provide a similar clinical insight into glucose homeostasis.

Regarding the patient’s preparation to perform the OGTT and MMTT, there are no considerable differences. The tests are both performed after an 8 h overnight fast and patients are advised not to consume alcohol, caffeine, or tobacco and to abstain from vigorous exercise since these factors can influence insulin sensitivity. Besides, around 100 g to 150 g of carbohydrates should be consumed daily for three days before the test since the dietary restriction of this macronutrient may impair glucose tolerance [26]. Glucose tolerance can also be influenced by other macro and micronutrients, for example, the percentage of energy resulting from fat and protein. However, the effect of these other macronutrients on glucose tolerance is not as immediate as in the case of carbohydrates. Therefore, it is not necessary to make changes in the amounts of fat and protein ingested in the days before the tests [35].

## 5. Mixed Meal Tolerance Test as a Method to Screen Glucose Disturbances

As previously mentioned, the MMTT provides a more physiological stimulus to insulin secretion. From a clinical perspective, performing this test could present some advantages in the screening of glucose and insulin disorders. Patients would also benefit from performing a more pleasant and palatable test with potentially fewer side effects and significantly less discomfort.

Some authors have already conducted clinical studies to compare the metabolic responses of OGTT to the MMTT and some of the main results of these studies are presented in Table 1. The correlations between the 2-glucose values for both tests were classified according to Evans (1996) [36].

In a cross-over study conducted by Chanprasertpinyo et al. [37], the authors assessed the effectiveness of using ice cream (described in Table 2) as an alternative to the 75 g oral glucose solution to diagnose diabetes and IGT in 104 healthy participants with no previous history of diabetes. The 2-h plasma glucose values were lower after the ice cream test compared with the OGTT (97.52 ± 40.71 mg/dL and 110 ± 55.53 mg/dL, respectively), but without statistical significance (Table 1). The discordance rate between the two tests was 9.61% when using the 2-h glucose values as the diagnostic criteria for diabetes. By combining the FPG values with the 2-h plasma glucose levels, the ice cream test would have missed 5.76% of the participants with a high risk of having diabetes. Two (1.9%) participants had hypoglycemia (38 and 49 mg/dL) after completing the OGTT, even though they did not demonstrate any symptoms. The authors also assessed the preferences and side-effects of the participants using a questionnaire and, according to this, 63% preferred the ice cream test and 36% reported more unpleasant symptoms with the OGTT.

Wolever et al. [38]’s results showed that the 2-h plasma glucose coefficient of variation of the mixed meal composed of wafers (described in Table 2) and the 75 g OGTT was not significant in people with diabetes. The plasma glucose levels 2-h following the mixed meal were closely associated with the ones 2-h after the OGTT, showing a linear correlation.

Marena et al. [39] compared the effects of a standard mixed meal (described in Table 2) and an OGTT on plasma glucose values on 40 participants equally distributed as healthy participants, IGT patients, mild non-insulin-dependent patients, and non-insulin-dependent with secondary failure to oral agent treatment. The levels of plasma glucose following the OGTT were significantly higher compared with the mixed meal in all groups. After the mixed meal, there was a statically significant difference in the mean 2-h plasma glucose levels among the four groups of participants. Regarding the plasma glucose incremental areas, these were significantly higher following the OGTT compared with the mixed meal, with exception to the healthy group. A significant correlation was found between plasma glucose incremental areas following the OGTT and the mixed meal (r = 0.511, *p* < 0.001). A highly significant (r = 0.956, *p*< 0.001) correlation between glucose values during the OGTT and mixed meal was also found. The values of glucose variation following the mixed meal were identical to the ones after the OGTT for the group with healthy participants.

The study conducted by Meier et al. [14], in a group of 60 participants, including 16 (26.7%) patients with T2DM, found a strong correlation between the 2-h glucose values after the 75 g OGTT and the corresponding glucose values following the test meal (described in Table 2) (r^2^ = 0.78, *p* < 0.0001). Additionally, peak glucose excursions following the glucose load were strongly associated with the correspondent maximum glucose values after the mixed meal administration [14]. Besides, there were statically significant correlations among the 120 min glucose values following the OGTT and the glucose values measured at all time points during the test meal (every 30 min until reaching 240 min).

Harano et al. [40] developed a cookie (described in Table 2) test to determine glucose, insulin, and lipids disturbances. The mean glucose and insulin values at 0, 1, and 2 h after the cookie test and the 75 g OGTT, performed on 19 healthy participants, were not statistically different. However, in two (11%) participants insulin response was earlier and higher in amplitude with the oral glucose solution. Similar to other studies, the 2-h glucose value with OGTT tended to be lower compared with the cookie test. According to the WHO criteria for evaluation of diabetes and IGT, 18 (94.7%) participants were classified as normal and one (5.3%) participant with IGT with the cookie test, whereas with oral glucose solution all 19 participants had normal glucose values. Reactive hypoglycemia was noted in five (26.3%) participants with the OGTT, and one (5.3%) with the cookie test. Participants reported epigastric symptoms only with the oral glucose solution.

This cookie test was also performed on a subgroup of 64 participants with lifestyle diseases and a subgroup of 26 participants that acted as the control for the lifestyle diseases subgroup. In the participants with diabetes (*n* = 14) (21.9%) from the subgroup with lifestyle diseases, glucose values were above 138.6 mg/dL following the cookie test. The insulin area under the curve (AUC) after the cookie test was used to measure insulin resistance and it showed that the participants with obesity (*n* = 24) (37.5%) and IGT (*n* = 6) (9.4%) had significantly higher values with the cookie test [40].

Traub et al. [41] compared the effectiveness of a muffin test to a 75 g OGTT in diagnosing IGT. Participants were 73 healthy adult women in the early postmenopausal stage. After an overnight 10-h fast, participants consumed a commercially available muffin (described in Table 2) and a coffee or tea with insignificant carbohydrate content. A subgroup of 12 (16%) participants performed the OGTT at a separate visit and an MMTT on a third visit. To perform the MMTT, the 12 participants were stabilized for 120 min before they ingested an oral shake with 600 kcal (described in Table 2). Compared with the MMTT and the OGTT, the muffin test worked well as a screening method to detect abnormal glucose metabolism. The mean 2-h glucose levels were significantly lower after the OGTT compared with the MMTT (*p* = 0.0002) and after the muffin test, but without being statistically significant (*p* > 0.05). Two (17%) participants of the subgroup were classified with IGT based on the muffin test but only one of these two women was identified in the OGTT. However, it should be noticed that the second participant had altered fasting glucose in the OGTT. Out of the 73 participants, eight (11%) were identified with IGT by the 2-h muffin test glucose. The screening of impaired glucose metabolism using the FPG cut-off (less than 126 mg/dL) would have a prevalence of missed cases of 63% (five in eight women). The data analysis demonstrated a significant correlation between the mean FPG and the 2-h muffin test glucose. These results were similar to the ones from a study by Freeman and colleagues [42]. They chose a meal that combined a commercially accessible muffin and orange juice (described in Table 2) to perform the MMTT in 13 women with polycystic ovarian syndrome (PCOS) since postprandial values may give a better representation of day-to-day glucose and insulin variations in these women. Only women diagnosed with PCOS without diabetes were included. In this study, the authors did not perform a statistical comparison between the results of the OGTT and the MMTT since the timing of the blood sample collection was different in the two tests (every 30 min in the MMTT and hourly in the OGTT). Only eight (62%) women performed the OGTT, thus having glucose measured with both the MMTT and the OGTT, and for one of these eight women insulin levels were not assessed during the OGTT. At the timepoints that coincided (60 and 120 min), glucose levels were lower in the MMTT in comparison with the OGTT (60 min: 111.40 ± 35.24 mg/dL and 142.57 ± 34.87 mg/dL; 120 min: 106.6 ± 30.78 mg/dL and 112.43 ± 25.79 mg/dL, respectively) and insulin levels were higher in the MMTT in comparison with the OGTT (60 min: 169.8 ± 132.9 µIU/mL and 154.62 ± 81.76 µIU7mL; 120 min: 152.05 ± 120.37 µIU/mL and 118.50 ± 97.34 µIU/mL, respectively). According to the cut-offs defined by ADA in 2008 [48], one (12%) woman showed IGT on both tests, one (12%) woman had IFG only on the OGTT, and one (12%) woman was classified as to have diabetes according to the results of the MMTT, but the values obtained in the OGTT were within the normal range. Out of the seven women that performed both tests, four (57%) had IGT on both the OGTT and MMTT [42]. No adverse effects were reported by the participants during and after the MMTT.

In a more recent study, Forbes et al. [43] studied 13 insulin-independent islet transplant patients. Twelve (92%) of the patients performed both tests, namely, the MMTT and the 75 g OGTT. In the MMTT a drink was administered, Ensure HP (Table 2), according to the participant’s body weight. Although there was no statistical significance, the data analysis showed a correlation between the glucose levels after the OGTT and MMTT. The results also demonstrated an association between the 90-min glucose ≥ 144.0 mg/dL following the MMTT and the 120-min OGTT cut-off point for diabetes diagnosis.

## 6. Meal Tolerance Tests to Screen for Gestational Diabetes Mellitus

Some authors have also been assessing the efficacy of meal tests in screening GDM and the main results of the selected studies are presented in Table 3. Currently, to diagnose GDM there are two possible approaches [26]: the one-step 75 g OGTT derived from the International Association of the Diabetes and Pregnancy Study Groups, where glucose is measured at a fasted state, 1-h and 2-h after the glucose load; and the two-step approach with a first screen, the glucose challenge test (GCT), using a 50 g glucose solution (nonfasting) and the assessment of 1-h plasma glucose value, followed by the second step, which is performing a 3-h 100 g OGTT for women who screen positive according to the Carpenter and Coustan criteria [26,49]. The summary of the main results is presented in Table 3.

Eslamian and Ramezani [50] also tested a standard breakfast containing 50 g of glucose, the equivalent of the first step to screen for GDM in the two-step approach. The 141 pregnant women enrolled in this randomized cross-over study performed both the 50-g glucose breakfast study and the standard 50-g GCT. Women with blood glucose values above 130 mg/dL after 60 min performed the 100 g 3-h OGTT to confirm the diagnosis. The breakfast test results were positive for 28 women (20%) while the GCT positively screened 41 women (29.3%). After performing the OGTT, it was confirmed that only 12 women (8.57%) had GDM. However, it was not reported how many of these pregnant women had been positively screened with the breakfast test. The concordance between the OGTT and GCT and breakfast test were 0.429 and 0.432, respectively (*p* < 0.001, Kappa test). The authors highlighted that the breakfast was tolerated by all the participants while the 50-g GCT was not tolerated by three women.

In the study by Marais et al. [44] conducted on 51 pregnant women, the 75-g OGTT with venous blood sampling diagnosed GDM in five (10%) patients, while the 75-g OGTT with capillary blood sampling diagnosed GDM in six (12%) women. The design breakfast glucose profile (DBGP) (described in Table 2) with capillary blood sampling diagnosed GDM in seven (14%) patients. The correlations between fasting capillary OGTT and DBGP (r = 0.639), 2-h capillary OGTT and DBGP (r = 0.542), fasting venous OGTT and fasting capillary DBGP (r = 0.608), and 2-h venous OGTT and 2-h capillary DBGP (r = 0.598) were statistically significant (*p* < 0.001). Comparing the diagnostic capacity of the gold standard, the DBGP test missed three (6%) women with GDM and gave five (10%) false-positive results, though three (6%) of these cases were due to higher values of fasting capillary glucose.

Coustan et al. [47] compared the plasma glucose values determined after a GCT with the value measured 1 h after a 600-kcal MMTT (Table 2) on 70 pregnant women, 20 (29%) of which were already diagnosed with GDM. In the subgroup of 50 pregnant women with presumed normal glucose, 24 (48%) performed the 100-g OGTT because one or both 50 g screening tests exceeded 130 mg/dL and four (8%) were diagnosed with GDM. For a cut-point of ≥120 mg/dL, out of the 20 women in the group with known GDM, 16 (80%) women were screened positive with the MMTT. Including the 24 cases of GDM, and for a threshold of ≥120 mg/dL, the sensitivity of the MMTT was 75% and the specificity was 94%.

Racusin and colleagues [45] tested strawberry candy twists (described in Table 2) as an alternative to the 50-g glucose solution (glucola) used in the GCT. The authors set the value to screen-positive for GDM as blood glucose equal to or above 130 mg/dL 1-h following the candy twists to ensure maximum detection of GDM. The two (10%) women diagnosed with GDM with the 3-h 100 g OGTT according to the National Diabetes Data Group criteria and the Carpenter and Coustan criteria [49,51] were screened as positive with the candy twists and the glucola drink. The candy twists screening would have prevented nine (45%) women from performing the 3-h OGTT. The results showed that the positive predictive value of the candy twists was higher than the glucola drink. However, only 11 (55%) of the 20 participants screened positive with the candy twists when all the participants had previously screened positive with the 50-g glucose beverage.

In the cross-over study conducted by Roberts et al. [46], 102 non-diabetic pregnant women performed both a 75-g OGTT and a standardized breakfast test (described in Table 2) on separated days within one week. Plasma glucose values were measured before and 2 h after the standardized meal. The correlation between the OGTT and breakfast test glucose values at 1-h (r = 0.36) and 2-h (r = 0.15) was weak. The authors used the WHO recommendations for diabetes diagnosis [52] from 1979 and identified seven (7%) women with IGT, established with a cut-off value of 144 mg/dL after the OGTT. However, when using the cut-off value of 162 mg/dL recommended by the Diabetic Pregnancy Study Group [53], only two (2%) women had IGT. Regarding the 2-h glucose values after the breakfast test, no women had values above 144 mg/dL. No women were diagnosed with GDM according to the 2-h OGTT glucose above 198 mg/dL.

In this same article [46], the authors also studied a group of pregnant women with IGT according to the WHO criteria [52]. Similarly, they did not find a significant correlation between the 2-h glucose values after the OGTT and the breakfast test (r = 0.35). Roberts and colleagues compared the 2-h glucose values after the OGTT with the highest glucose values after the breakfast test, and out of the 104 pregnant women with IGT, only 15 (14%) had values above 144 mg/dL.

## 7. Discussion

Some studies suggest that the MMTT is more similar to the physiological postprandial metabolic responses compared with the OGTT. In this review, we analyzed the results from 13 studies, five of them conducted in pregnant women, that compared the efficacy of an MMTT in screening or diagnosing DM and GDM. Half of the studies not performed in the context of GDM showed a strong or very strong correlation between the 2-h glucose values of both tests [14,37,38,39] and low percentages of missed or misdiagnosed cases [37,40,41,42], which may indicate that the MMTT could serve as an alternative to the OGTT. As for GDM, the results showed that a high number of women screened positive for GDM using the MMTT without having that condition, as proven by the low percentage of PPV and sensitivity [45,50,54]. The results are promising, and in the future, the MMTT could be a valid alternative to the OGTT. However, considering the current data and options available, it is not yet possible to make this transition. There are still some key points that should be taken into consideration, including the definition of cut-off points to screen and diagnose diabetes with the MMTT and the standardization of a mixed meal.

When comparing both tests, gastric emptying is a factor that should be considered since it is quicker after the OGTT compared with a mixed meal, which leads to a fast release of glucose to the duodenum and the portal venous circulation [55,56,57]. Additionally, the protein and fat content present in the mixed meals delay gastrointestinal glucose absorption, which leads to lower profiles of postprandial glucose and different insulin secretion profiles [58,59]. The types of mixed meals used in the tolerance tests are also, generally, more palatable and acceptable than the OGTT, causing fewer side effects such as stomach discomfort, headache, dizziness, hunger, and nausea [38]. The high glucose content of the solution used in the OGTT leads to a fast increase in blood glucose and, consequently, in most cases, a fast rise in insulin values, which next provokes a sudden decline in blood glucose with dizziness and syncope risk. Reactive hypoglycemia is one of the unpleasant side effects of this test and, according to the study by Harano and colleagues [40], it was more prevalent in this test than in the meal tolerance test. Preventing these symptoms and side effects can be particularly important in pregnant women, who are usually more sensitive to this type of test. These solutions could also prevent the need to repeat the OGTT in pregnant women due to the vomiting induced by the glucose solution, as occurred in the study by Roberts and colleagues [46]. According to the results of the study conducted by Forbes and colleagues [43], the MMTT could even reduce the time necessary to perform a diagnostic test. In this study, the 90-min glucose values after the MMTT were associated with the 120-min OGTT cut-off point for the diagnosis of diabetes, which could spare 30 min in test time.

The administration of a standard mixed meal as the challenge test for screening diabetes and impaired glucose tolerance may be an alternative to the glucose load, providing that the standardized meal has an expected effect on glucose levels [47]. Brodovicz et al. [10] suggested that a liquid meal test could also be another solution to explore since it may be easier to standardize and administer. In their study, postprandial metabolic responses were very similar and well-correlated after a mixed or liquid meal test with comparable nutritional and energetic value; however, the participants did not perform an OGTT to compare the metabolic responses. Although, when looking for a more physiological stimulus to diagnose impaired glucose metabolism, the liquid meal approaches an OGTT in the sense that it is not as physiological as a solid mixed meal due to the more rapid delivery of nutrients to the duodenum. It has, however, the advantage of avoiding the confounding factor of delayed gastric emptying [43,60].

The non-completion of OGTT delays the diagnosis of diabetes, which increases the risk of developing complications associated with this condition. One of the main reasons cited for non-completion of the OGTT by pregnant women is related to the fact that they cannot tolerate the test procedure [61]. The results of the study by Roberts and colleagues [46] showed that pregnant women that performed the OGTT first had a higher withdrawal rate, which can be an indicator that this procedure was less tolerable. Investigating new methods and/or alternatives for the glucose solution used in the diagnosis of diabetes, which are easier and more tolerated, has significant clinical importance but also shortcomings. Though the mixed meal tolerance tests can be recognized as a substitute test for postprandial responses, comparisons of metabolic responses to the MMTT among studies cannot be made due to the diversity of meal sizes and nutritional contents that have a different impact on gastric distension, gastric emptying, and incretin response [59,62,63,64,65]. An MMTT performed with a standard mixed meal would simplify comparisons of results across countries and studies, reduce the variability, and contribute to the validation of this methodology. The standardization of the meals used to screen for diabetes was previously emphasized by Marais et al. [54]. They performed a study on 50 pregnant women to measure the carbohydrate quantity of non-standardized breakfast meal tests used to screen GDM. Although the carbohydrate content median was 71 g, similar to the 75 g oral glucose solution, the values ranged from 55 g to 145 g. Though the standardized meal could represent a substitute for the oral glucose solution, the authors advise against using non-standardized meals since only seven of the meals (14%) fell within the 10% of 75 g carbohydrate target [54]. Besides, similarly to the OGTT, to guarantee homogeneity in the meal test is necessary to ensure that the patients ingest the food in its entirety and in an equal interval to minimize the effect of confounding factors. In the case of the study by Harano et al. [40], participants had 10 to 15 min to eat the cookie, but if they were used to a small breakfast or if they did not like the cookie, they could eat half of it within 10 min and the other half within an additional 10 to 20 min.

Regarding costs, the meal tests are, in general, more affordable than the oral glucose solution. Some authors set the cost of the oral glucose solution at $6 per unit [40], others at $5.20 [41], while others just mention that the costs range from $3.41 to a lot higher [45]. Besides, a recent literature review emphasized the need to perform more studies comparing the composition and homogeneity of different oral glucose solutions since there is only a small number of studies and most of them are not appropriately designed [66]. The composition of these solutions, including the components added to improve organoleptic characteristics such as taste and smell, may influence blood glucose and endogenous insulin secretion.

In addition to being more equivalent to people’s dietary patterns, the meal tolerance tests can provide valuable information concerning the different categories of glucose intolerance and insulin resistance. Though, more research is needed to define the threshold of blood glucose values when using the MMTT as a screen or diagnosis method for diabetes or other glucose intolerances. In the study by Coustan et al. [47], when the threshold was lowered to 100 mg/dL, the sensitivity increased from 75% to 96%; however, the specificity lowered from 94% to 74%. To screen for glucose disturbances, the definition of the cut-off point is of the utmost importance to avoid the administration of unnecessary OGTT, but with this value, the healthcare professionals need to be able to detect people with middle abnormal glucose tolerance tests.

Some limitations were identified while conducting this review that may be important to consider in the design of future studies. The studies included in this review presented some heterogeneity regarding the tests performed and the screening and diagnostic criteria selected by the authors. The differences in both quantity and nutritional values of the food items administered in the diverse MMTT can influence the metabolic responses of the participants [29,30]. Besides, some of the studies included herein compared the MMTT to a 75 g [14,37,38,39,40,41,42,43,46] or 100 g glucose load [45,47,50], while others compared it to a GCT [47,50], and one study used capillary glucose values [44] instead of blood glucose. A few of the studies included were performed more than twenty years ago; thus, some of the methodologies may be outdated (i.e., Roberts et al. [46] assessed IGT in pregnant women using the WHO criteria from 1980, which are currently outdated). Besides, none of the articles explained the process behind the formulation or choice of the meals tested, and some of them did not provide the nutritional information of the meal.

Future research in the area can address the limitations of previous studies and contribute with more information such as the acceptability and palatability of the alternative tests to the participants.

Currently, there is still a lack of clinical studies testing this hypothesis of a mixed meal as an equivalent to the glucose solution. To validate the MMTT as an alternative to the OGTT for a specific population, there is a need to design and perform more multicenter randomized clinical trials with larger sample sizes. However, designing a trial to assess and compare the diagnostic accuracy of an MMTT with the OGTT may be challenging due to the sensitivity and specificity of the OGTT being equivalent to 100%. Thus, to demonstrate a greater accuracy and precision in diagnosis, it is still necessary to show a better degree of concordance among repeated tests in clinical studies with a much larger number of participants than those present in this article.

## 8. Conclusions

A complete nutritional challenge that incorporates all the macronutrients (carbohydrates, proteins, and fat) such as the mixed meal tolerance tests is theoretically more physiological. Therefore, this type of test is likely to provide more insightful and comprehensive information concerning metabolic and glucose homeostasis compared with a single macronutrient challenge. However, the diversity of mixed meal challenges already tested highlights the need to perform larger trials to compare the effectiveness in diagnosing glucose disturbances with a standardized mixed meal and the oral glucose solution administered in the OGTT. The clinical application of the MMTT as a validated test to diagnose glucose disturbances in patients warrants further investigation.

## Figures and Tables

**Table 1 nutrients-14-02032-t001:** Characteristics and main results of the selected studies comparing the metabolic effects of the oral glucose tolerance test to the meal tolerance tests.

Ref.	Participants	Main Results	2-h Glucose Correlations
**Randomized cross-over studies**			
Chanprasertpinyo et al. 2017 [37]	Healthy adults without DM (*n* = 104) 30 M; 74 F	**2-h glucose levels (OGTT/ice cream * test)**: ρ = 0.82; *p* < 0.001; 9.61% discordant diagnostic results **Ice cream test**: 5.76% of missed DM cases	+ Very strong
Wolever et al. 1998 [38]	Adults with normal weight, obesity, IGT, or diabetes (*n* = 36) 15 M; 21 F	**2-h glucose levels (OGTT/MMTT):** r = 0.97) **1-h glucose (MMTT)/2-h glucose (OGTT)**: r = 0.96 **1-h glucose (OGTT)/2-h glucose (MMTT)**: r = 0.91	+ Very strong
Marena et al. 1992 [39]	Adults with NGT, IGT, mild NIDDM, or NIDDM (*n* = 40; 10 by group) 20 M; 20 F	**Glucose incremental areas (OGTT/mixed meal *)**: r = 0.511, *p* < 0.001 **2-h glucose values (OGTT/mixed meal)**: r = 0.956, *p* < 0.001	+ Very strong
**Non-randomized cross-over studies**			
Meier et al. 2009 [14]	Adults with NGT, IGT, or diabetes (*n* = 60)	**2-h glucose levels (OGTT/MMTT)**: r^2^ = 0.78, *p* < 0.0001	+ Strong
Harano et al. 2006 [40]	Healthy adults (*n* = 19) 6 M; 13 F	**Cookie * test**: 1 (5%) IGT **OGTT:** 19 (100%) normal blood glucose	n.s.
**Cross-sectional studies**			
Traub et al. 2012 [41]	Healthy early postmenopausal women (*n* = 12)	**MMTT**: 1 (8%) of 2 (16%) participants identified with IGT (confirmed with the OGTT). The second participant had abnormal fasting glucose with the OGTT	n.s.
Freeman et al. 2010 [42]	Women with PCOS (*n* = 8)	**Blood glucose levels** **OGTT and MMTT**: 1 (12%) IGT **OGTT**: 1 (12%) IFG (not with MMTT) **MMTT**: 1 (12%) diabetes (not with OGTT) **Blood insulin levels** **OGTT and MMTT**: 4 (57%) IGT	n.s.
**Retrospective study**			
Forbes et al. 2018 [43]	Adults with T1DM and stable transplant grafts (*n* = 13) 9 M; 4 F	**2-h glucose values (OGTT and MMTT *)**: r = 0.45; *p* = 0.07 90-min MMTT glucose ≥ 144 mg/dL: equivalent to 2-h OGTT glucose ≥ 199.8 mg/dL	+ Moderate

* Nutritional composition described in Table 2. DM, Diabetes Mellitus; M, Male; F, Female; OGTT, Oral Glucose Tolerance Test; IGT, Impaired Glucose Tolerance; NGT, Normal Glucose Tolerance; NIDDM, Non-Insulin-Dependent Diabetes Mellitus; MMTT, Mixed Meal Tolerance Test; PCOS, Polycystic Ovarian Syndrome; T1DM, Type 1 Diabetes Mellitus. n.s., not stated.

**Table 2 nutrients-14-02032-t002:** Nutritional characteristics of the products and/or meals tested.

Ref.	Product	Energy (kcal)	Carbohydrates (% TE, g)	Protein (% TE, g)	Fat (% TE, g)
Forbes et al., 2018 [43]	Ensure HP	1.1 kcal/mL	55%	22%	23%
Marais et al., 2018 [44]	Future Life Excel meal	n.d.	75.0 g	n.d.	n.d.
Chanprasertpinyo et al., 2017 [37]	Ice cream	620.9	73.9 g	18.9 g	27.7 g
Racusin et al., 2015 [45]	10 strawberry-flavored candy twists (Twizzlers)	n.d.	50.0 g (91.968%)	3.515%	4.527%
Traub et al., 2012 [41]	Muffin (Beigel’s Bakery)	410.0	56.0 g	6.0 g	18.0 g
Traub et al., 2012 [41]	Shake	600.0	75.0 g	30.0 g	20.0 g
Freeman et al., 2010 [42]	Muffin (Costco) and orange juice (Tropicana)	800.0	105.0 g	12.0 g	38.0 g
Meier et al., 2009 [14]	Continental breakfast ^1^	820.0	90.0 g	26.8 g	39.2 g
Harano et al., 2006 [40]	Cookie	533.0	75.0 g	7.0 g	25.0 g
Wolever et al., 1998 [38]	5 wafers (DSP)	345.0	50.0 g	12.1 g	10.7 g
Roberts et al., 1997 [46]	Standardized breakfast ^2^	300.0	45.0 g	10.0 g	9.0 g
Marena et al., 1992 [39]	Standard mixed meal ^3^	590.0	69.0 g (44.0%)	22.6 g (15.0%)	27.0 g (41.0%)
Coustan et al., 1987 [47]	Standard test breakfast ^4^	600.0	52.0 g	28.0 g	31.0 g

TE, total energy; n.d., not defined: DSP, diabetes screening product.^1^ Two European bread rolls; 20 g of butter; 40 g of gouda cheese; 30 g of jam; one egg; 150 g of yogurt; 200 mL of tea.^2^ Breakfast cereal with milk; toast and butter; tea (amounts not expressed).^3^ 125 g of fruit juice; 75 g of ham; 89 g of white bread.^4^ Two scrambled eggs; two slices of toast or one English muffin; two pats of butter or margarine; 8 oz orange juice; 8 oz whole or skim milk; one cup of coffee or tea (no sugar).

**Table 3 nutrients-14-02032-t003:** Main characteristics and results of the selected studies that used the meal tests to screen gestational diabetes mellitus.

Ref.	Participants	Main Results	2-h Glucose Correlations
**Case-control study**			
Eslamian and Ramezani 2006 [50]	Pregnant women (*n* = 141)	**GCT:** 41 (29.3%) GDM **OGTT:** 12 (8.57%) GDM **Breakfast test:** 28 (20%) GDM Optimal cut-off value: 130 mg/dL at 60 min (83.3% sensitivity; 85.9% specificity; 35.7% PPV; 98.2% NPV)	n.s.
**Randomized cross-over studies**			
Marais et al., 2018 [44]	Pregnant women with a high risk of GDM (*n* = 51)	**2-h OGTT (venous):** 5 (10%) GDM **2-h OGTT (capillary):** 6 (12%) GDM **2-h DBGP test (capillary):** 7 (14%) GDM; 3 (6%) missed GDM cases; 5 (10%) false-positive cases **DBGP test:** 25% sensitivity; 96% specificity; 33% PPV; 95% NPV	n.s.
Coustan et al., 1987 [47]	Pregnant women with GDM (*n* = 20)	16 (80%) of the 20 subjects with GDM had a 1-h breakfast test plasma glucose level ≥ 120 mg/dL (threshold defined by the 1-h mean glucose + 2 SD) **MMTT:** 75% sensitivity; 94% specificity	n.s.
**Non-randomized cross-over studies**			
Racusin et al., 2015 [45]	Pregnant women screened positive for GDM (*n* = 20)	**1-h candy twists test:** 100% sensitivity; 50% specificity; 18% PPV; 100% NPV; 82% false-referral rate; 18% detection rate	+ Moderate **
Roberts et al., 1997 [46]	Non-diabetic pregnant women (*n* = 102)	**OGTT (cut-off 144 mg/dL):** 7 (7%) IGT **Breakfast (cut-off 144 mg/dL):** 0 (0%) IGT **OGTT (cut-off 162 mg/dL):** 2 (2%) IGT **OGTT (cut off 192 mg/dL):** 0 (0%) GDM	n.s.

GDM, Gestational Diabetes Mellitus; OGTT, Oral Glucose Tolerance Test; GCT, Glucose Challenge Test; PPV, Positive Predictive Value; NPV, Negative Predictive Value; DBGP, Design Breakfast Glucose Profile; SD, standard deviation; MMTT, Mixed Meal Tolerance Test; IGT, Impaired Glucose Tolerance. n.s., not stated; ** *p* < 0.05.

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
