# Peer review of "Metabolic Effects of an Oral Glucose Tolerance Test Compared to the Mixed Meal Tolerance Tests: A Narrative Review"

_nutrients, 2022, doi:10.3390/nu14102032_

Round 1
Reviewer 1 Report
This is a narrative review of studies that have directly compared the metabolic effects (glucose and insulin responses, ability to identify abnormalities in glucose homeostasis, etc.) of mixed meal tolerance tests (MMTTs) with the oral glucose tolerance test (OGTT). While the topic is important, and the paper includes potentially useful information for clinicians and researchers, I found the paper to be a bit disorganized. It also lacks a synthesis and overall summary of the collected data. Please see specific comments below.
- Lines 56-57: suggest that a more detailed explanation should be provided about the “re-analytical, analytical, and post-analytical variables” that interfere with the use of the OGTT for diagnosis of gestational diabetes mellitus
- Lines 67-68: suggest that it be made clear that the “topic” referred to is studies that directly compared OGTT and mixed meals tolerance tests
- Lines 76-78: suggest clarification of the exclusion criterion that says “studies were not carried out within the scope of the research”; also, since this was a narrative review, why were review articles excluded? It seems that an evaluation of any prior review papers on this topic would be informative for a narrative review
- Lines 80-81: because this is a narrative review and not a systematic review, suggest that clarification may be needed about what is meant by “the information was analysed and synthesized” from the selected articles.
- Suggest that Sections 2.1 and 2.2 in the Materials and Methods are not appropriately placed here as these sections simply cover topics relevant to the review, i.e., methods for diagnosing diabetes and methods of the OGTT and MMTT, not methods for collecting papers for this review; suggest moving these from sub-headings under the Methods, to stand-alone sections
- Lines 152-153: suggest re-wording for clarification the sentence, “Though, these are longer-term influences, so, there is not a need to do changes before the tests (37).”
- Table 1, General comment: suggest splitting the studies reported in Table 1 into 2 separate tables according to whether the trials were cross-sectional or interventional studies, or perhaps the table could be organized with separate sections for each rather than intermingling them. This re-organization would apply to the description of the studies in the text also.
- Table 1, suggest adding to the table title that these are “selected” studies, since this was not a systematic review and it is possible that other studies have been conducted (particularly since only 1 database was used in the literature search)
- Table 1: it is unclear what is meant by “nonmetabolic participants” and “metabolic participants” in the description of the Traub et al. (40) study
- General comment: in the text when results are reported as numbers of subjects, suggest also reporting the % of the population that these represent; also suggest providing results for glucose, insulin, etc. in consistent units where possible, i.e., mg/dL or mmol/L, instead of a mixture of both
- Table 1: in the description of the Freeman et al. (41) study, the key findings are not entirely clear; why are there glucose results for just 8 subjects (and 7 subjects for insulin) when there were apparently 13 subjects in the study?
- Table 1, General comment: suggest streamlining the detailed descriptions of the participants in the table, especially since the main results are often not presented according to the various subgroups in this table; also suggest expanding the Main Results in the table with more details for clarification
- Table 1: it is unclear what is mean by “under daily life conditions” in the description of the Meier et al. (16) study
- Table 1: the study design description of cross-sectional and crossover is unclear for the Harano et al. (42) study
- Beginning at Line 182, the description of the Traub et al. (40) study is a bit confusing. For example, in Line 187, it is unclear what is meant by “In this last test”; also rather than stating “According to the authors…the muffin test worked well as a screening method…” that your own evaluation of the comparison between OGTT and MMTT in this trial should be provided
- Table 2: suggest removing the column for cost, as this is expected to be highly variable, and was not defined for most of the products anyway
- Table 2, Footnote 2: are there amounts available for the cereal, milk, toast, butter, etc. for the standardized breakfast?
- Line 310: the name “glucola” has not been introduced prior to this point; suggest explaining it
- Line 294: the description of the study procedures of the Eslamian (51) study is not clear; it says that it tested a standard breakfast containing 50-g glucose, but then later it appears that the breakfast study was different from the 50-g glucose challenge; suggest clarifying what was done
- Line 316: the description of the Roberts et al. (48) study is quite short compared to all of the others – just 1 sentence compared with full paragraphs for the others
- Table 3: same comment as for Table 1, suggest describing the participants in a more general manner without as many details
- Table 3, in the description of the Racusin et al. (47) study, it says “double-crossover design” should this instead be “double-blind, crossover design”?
- Discussion: suggest that the paper is missing a synthesis of the evidence, i.e., what is the take-home message? Did this review indicate that the MMTT is similar “enough” to the gold-standard OGTT in identifying abnormalities in glucose tolerance in order to be useful clinically and/or in research? Suggest that this summary is needed as the 1st paragraph of the Discussion.
Minor:
- Line 18: suggest rather than “being the mixed meal tolerance test” that this could say “such as occurs with a mixed meal tolerance test”
- Line 52: suggest rather than “does not enlighten on” that this could say “does not provide information about”
- Line 109: suggest that “time form diagnosis” should be “time from diagnosis”
- Line 174: suggest that “no previous story of diabetes” should be “no previous history of diabetes”
- Line 324: “with know GDM” should be “with known GDM”
Author Response
Please see the attachment with the replies to your comments and suggestions.
Thank you.

Reviewer 2 Report
Lages et al present a review of the advantages and disadvantages of using an oral glucose tolerance test (OGTT) compared to a mixed meal tolerance test (MMTT) in the diagnosis of diabetes & glucose intolerance. The authors thoroughly review the available literature and highlight the discrepancies & differences in the available studies.
Comments:
Overall, this review is well-written. The Tables are useful for quickly comparing similar & dissimilar data among the studies; however, either due to PDF formatting or size constraints the Tables are difficult to view & synthesizing information form them is difficult (particularly Table 1). Table 2 is much more concise and easy-to-read: is there information or criteria in Table 1 that can be expressed similarly as Table 2? For instance, if the study determined correlation between OGTT and MMTT, this can be expressed in a column, or marked N/A if not applicable to that study. The "Main Results" column of Table 1 could then be simplified to the most important points.
Author Response

(The authors gave the same response as above.)

Round 2
Reviewer 1 Report
Thank you for your attention to the previous suggestions for revisions. The additions to the 1st paragraph in the Discussion now provide a clear summary of the overall findings, and the simplification of Tables 1 and 3 improve the ability to understand the findings in each of the studies reviewed.